# Comparative genomics of methicillin-resistant and -susceptible *Staphylococcus aureus* from Mexico City hospitals

Irma Martínez-Flores,[1] Patricia Bustos,[1] Rosa I. Santamaría,[1] Alan Aguayo-González,[1] Eugenia Silva-Herzog,[2] Xavier Soberón,[3] Víctor González,[1] Roberto Cabrera-Contreras[4]

**ABSTRACT** Methicillin-resistant (MRSA) and methicillin-susceptible (MSSA) *Staphylococcus aureus* are major causes of hospital-acquired infections worldwide. However, their genomic features remain underexplored in many regions, particularly in low- and middle-income countries (LMICs). Here, we investigated the genomic diversity, antibiotic resistance, and virulence traits of methicillin-resistant (MRSA) and methicillin-susceptible (MSSA) *Staphylococcus aureus* isolates from three tertiary care hospitals in Mexico City. A total of 101 isolates collected between 2006 and 2019 from newborns and adults with diverse infections were analyzed using whole-genome sequencing. MRSA isolates were restricted to clonal complexes CC5 and CC8, whereas MSSA strains displayed a broader diversity across multiple lineages. A distinct clade of CC8-MRSA isolates identified in 2013 among newborns was closely related to the USA300 lineage but carried SCC*mec* IVc without the Panton-Valentine leukocidin genes and the COMER island. Similarly, a localized cluster of CC5-MSSA isolates from neonates in 2012 showed genomic variations largely driven by prophage-associated elements. MRSA strains carried a higher burden of resistance genes, including fluoroquinolone-associated mutations, whereas MSSA isolates exhibited greater heterogeneity in virulence genes and *spa* types. Both groups shared a conserved virulence gene repertoire; however, variations in the virulence genes highlighted lineage-specific pathogenic features.

**IMPORTANCE** MRSA and MSSA *S. aureus* genomes are still largely unstudied in Mexico although they are frequently found in hospitals. This study provides a long-term genomic analysis of MRSA and MSSA isolates from three hospitals in Mexico. Our findings revealed the presence of *S. aureus* international clones of clonal complexes CC5 and CC8 over a decade, broader diversity in MSSA, and localized clonal transmission within hospitals. The contrasting resistance and virulence profiles of MRSA and MSSA clones underscore the need for genomic surveillance frameworks to inform infection control and antibiotic stewardship in healthcare settings.

**KEYWORDS** *Staphylococcus aureus*, whole-genome sequencing, antibiotic resistance, virulence, comparative genomics, MRSA, MSSA, genotyping, clonal transmission

S taphylococcus aureus is an opportunistic pathogen responsible for a wide spectrum of infections, ranging from superficial skin lesions to life-threatening conditions such as pneumonia, bacteremia, and sepsis (1). Globally, two major clinical variants of *S. aureus* are distinguished by their susceptibility to β-lactam antibiotics: methicillin-resistant *S. aureus* (MRSA) and methicillin-susceptible *S. aureus* (MSSA). MRSA poses a major public health concern because of its resistance to multiple antibiotics and its frequent association with hospital-acquired infections, especially in immunocompromised patients and newborns (2). Likewise, community-associated MRSA (CA-MRSA)

**Peer Reviewer** Andrew David Berti, Wayne State University, Detroit, Michigan, USA

Address correspondence to Víctor González, vgonzal@ccg.unam.mx, or Roberto Cabrera-Contreras, rcc@unam.mx.

Irma Martínez-Flores and Patricia Bustos contributed equally to this article. Author order was determined by their experimental and bioinformatic contributions.

The authors declare no conflict of interest.

See the funding table on p. 14.

strains, often carrying SCC*mec* type IV and Panton–Valentine leukocidin (PVL) genes, have emerged as significant agents of severe infections outside the hospital setting (3–7)

MSSA strains, although susceptible to β-lactams, can cause diseases similar to those caused by MRSA. They are broadly distributed in both healthcare and community environments and can cause invasive and life-threatening infections (8–10). Despite this, MSSA has historically received less attention in genomic surveillance efforts than MRSA, limiting our understanding of its diversity, transmission dynamics, and potential virulence.

The population structure of *S. aureus* is predominantly clonal and shaped by selective pressures, such as antibiotics and horizontal gene transfer of diverse virulence genes (11, 12). While MRSA evolution is closely associated with the acquisition of the SCC*mec* element (13–15), both MRSA and MSSA lineages carry diverse repertoires of virulence factors, such as adhesins, biofilm-associated genes, toxins, and global regulators (1, 16–25). These genomic traits contribute to pathogen persistence, adaptability, and clinical outcomes.

Although extensive genomic studies have been conducted in Europe, North America, and parts of Asia to investigate the evolutionary dynamics of MRSA and MSSA, there is a notable lack of comprehensive genomic surveillance in Latin America, particularly in Mexico. Existing reports on *S. aureus* in Mexican hospitals have largely relied on traditional microbiology or targeted molecular markers and have provided only fragmented insights into the genomic structure, resistance, and virulence of circulating strains (26–29). This knowledge gap limits the development of tailored infection control strategies and our understanding of local transmission and clonal expansion.

To address this gap, we conducted a genomic analysis of 101 *S. aureus* isolates collected between 2006 and 2019 from three tertiary care hospitals in Mexico City. This data set includes isolates from diverse infections in neonates and adults, including catheter-related, wound, bloodstream, and osteoarticular infections. Specifically, we aimed to determine (i) the extent of the genetic diversity of *S. aureus* MSSA and MRSA and how it is structured across hospitals, (ii) the genomic differences in virulence and antibiotic resistance factors between MRSA and MSSA isolates, and (iii) assess evidence for clonal transmission across hospital environments.

## MATERIALS AND METHODS

### *S. aureus* hospital isolates

A total of 286 *S. aureus* isolates were obtained from a *Staphylococcus* strain collection maintained at the Faculty of Medicine, UNAM, as described previously (30). Isolates were obtained between 2006 and 2020 from three tertiary care hospitals in Mexico City: 187 from the National Institute of Perinatology (INPer, 2006–2016), 71 from the National Institute of Cardiology (INC, 2016–2018), and 28 from the Hospital de Traumatología y Ortopedia (HDF, 2019–2020). Isolates were obtained from diagnostic samples, including blood, wound and tissue secretions, catheters, and osteoarticular tissues (see Tables S1 and S2 at https://github.com/aguayo-alan/Martinez-Flores_et_al_Microbiology_Spectrum_2026). All isolates were confirmed as *S. aureus* using standard microbiological and molecular methods, including coagulase testing and PCR of the *mecA* gene.

A set of 101 isolates (39 MRSA and 62 MSSA) from the full collection of 286 isolates was used for whole-genome sequencing (Fig. 1; see Table S2 at https://github.com/aguayo-alan/Martinez-Flores_et_al_Microbiology_Spectrum_2026). The 101 isolates set composition criteria aimed to preserve the temporal distribution of MRSA and MSSA isolates and to ensure representation across hospitals and major clinical infection types (Fig. 1). While the overall MRSA/MSSA ratio and the inclusion of isolates from all hospitals and study years were maintained, the set was not fully representative at the level of specific infection sites. In particular, clinical sources such as blood and tissue secretions were overrepresented. Once these criteria were met, 101 isolates were randomly chosen

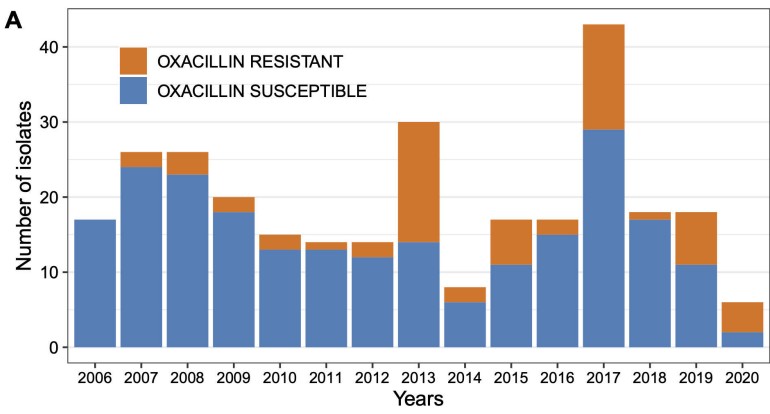

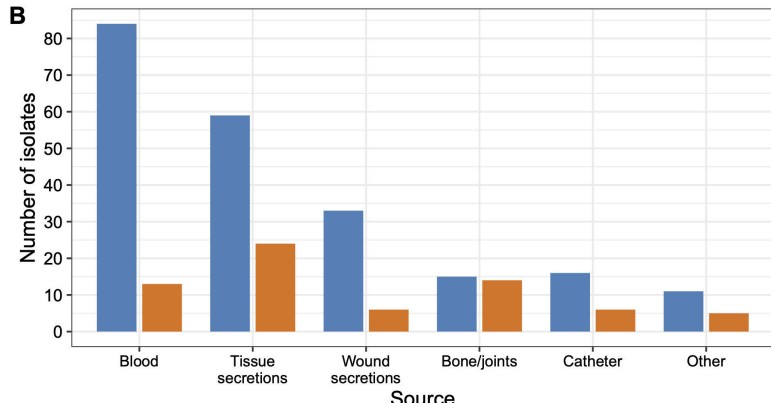

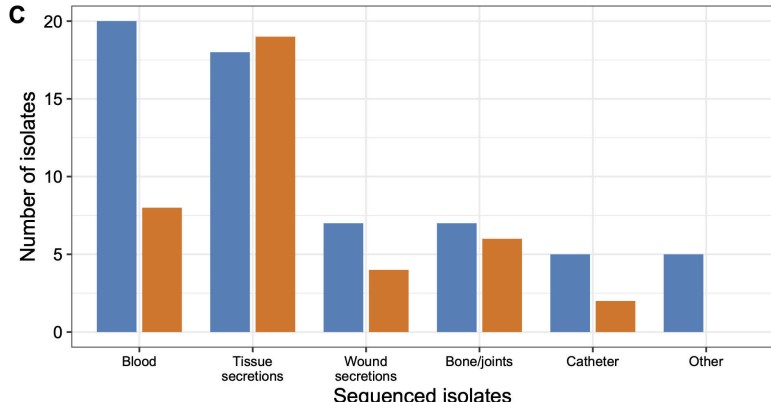

**FIG 1** Clinical origin of *Staphylococcus aureus* isolates. (A) Annual distribution of oxacillin-susceptible (blue) and resistant (orange) isolates ($n$ = 286) collected between 2006 and 2020 from three tertiary care hospitals in Mexico City. (B) Distribution of oxacillin-susceptible (blue) and resistant (orange) isolates by infection source in the full collection of 286 isolates. (C) Distribution of oxacillin-susceptible (blue) and resistant (orange) isolates by infection source in the 101 isolates used in the genomic study.

(per year, hospital, and infection source). Accordingly, the sequenced set was designed to support comparative genomic analyses rather than epidemiological inferences.

## Bacterial growth and characterization

*S. aureus* strains were cultivated and characterized as previously described (30). Briefly, primary isolates were grown and characterized in selective mannitol salt (Oxoid CM0085). For subsequent experiments, cultures were grown in lysogeny broth (LB) at

37°C with shaking at 210 rpm. BACTO agar was added when solid plates were required to culture the bacteria. *Staphylococcus* species were identified using VITEKR 2 equipment with a VITEKR 2 GP ID card, testing 64 properties of gram-positive bacteria (bioMérieux SA, 376 Chemin de l'Orme, France). Susceptibility to oxacillin was determined according to the Clinical and Laboratory Standards Institute (CLSI) Approved Standard M100 (https://clsi.org/) used a minimum inhibitory concentration (MIC) ≥0.4 µg/mL for oxacillin (CLSI Approved Standard M100) as described by the Centers for Disease Control and Prevention (https://www.cdc.gov/mrsa/about/index.html).

## Biofilm assay

Overnight cultures of bacteria in LB medium ($DO_{600}$ = 1.8) were diluted 1:100 in fresh medium, and 200 µL (approximately $1 \times 1.5 \times 10^7$ CFU/mL) of this dilution was added to the wells of a 96-well microplate (Corning 3595 Costar). The plates were incubated at 37°C for 24 h without shaking to promote biofilm formation. After incubation, the plates were washed three times with 200 µL sterile PBS to remove non-adherent bacteria. The remaining attached bacteria were fixed with 200 µL of 99% methanol for 15 min, removed, and dried for 10 min. The plates were stained with 200 µL of crystal violet (0.1%) for 10 min at room temperature. The crystal violet solution was removed, and the wells were washed thrice with distilled water and then allowed to dry at room temperature for 15 min. Finally, the dye bound to the adherent cells was resolubilized with 200 µL per well of glacial acetic acid (33%, vol/vol) for 15 min. $OD_{570}$ was measured using a microplate reader (Bio-Tek Synergy 2 Multi-Mode). Based on OD, bacterial cultures were classified as non-adherent, weakly adherent, moderately adherent, or strongly adherent using the criteria established by Stepanović et al. (31). The cut-off OD for this assay is defined as three standard deviations above the mean $OD_{570}$ of the negative control (0.16) (31). The assay was performed in triplicate.

## Genome sequencing

Genomic DNA was extracted using the GenElute Bacterial Genomic DNA Kit (Sigma-Aldrich) according to the manufacturer's protocol. Briefly, 1.5 mL of an overnight bacterial broth culture was collected and resuspended in 200 µL of lysozyme solution (lysostaphin 200 units/mL, Sigma Aldrich), and 20 µL of RNase A solution (20 mg/mL) was added to obtain RNA-free genomic DNA. The cells were lysed with 200 µL lysis solution C (B8803 Sigma-Aldrich) and 20 µL Proteinase K (20 mg/mL). The lysate was filtered through DNA-binding columns, washed, and eluted. DNA integrity and purity were assessed using agarose gel electrophoresis and spectrophotometry (NanoDrop 2000; Thermo Fisher Scientific). The final DNA concentration was assessed using a Qubit fluorometer dsDNA assay (Thermo Fisher Scientific). Genome sequencing was performed using BGI Tech Solution (BGI Americas Corporation, Cambridge, MA, USA) with DNA nanoball PCR-free whole-genome sequencing (DNBSEQ WGS). Libraries containing 350 bp inserts were sequenced, obtaining 2 Gb of 150 bases per read using a paired-end sequencing (PE) strategy. The raw data were then processed for quality control and cleaning using FastQC v0.11.8 (32) and Trim Galore v0.6.4 (https://github.com/FelixKrueger/TrimGalore).

## Genome assembly and annotation

High-quality sequence reads were assembled *de novo* using SPAdes genome assembler v3.13.1 (33), resulting in 101 draft genomes of approximately 2.8 Mb with a high coverage of 600–700×, a median number of contigs of 44, and an N50 higher than 285 kb (see Fig. S1 at https://github.com/aguayo-alan/Martinez-Flores_et_al_Microbiology_Spectrum_2026). MUMmer-based average nucleotide identity (ANIm) was used to determine the taxonomic classification by comparing each sequence with *S. aureus* reference genomes (see Fig. S2 and Table S3 at https://github.com/aguayo-alan/Martinez-Flores_et_al_Microbiology_Spectrum_2026) (34, 35). Genome annotations

were obtained from "The Bacterial and Viral Bioinformatics Resource Center"' (BV-BRC, https://www.bv-brc.org/). Annotation of antibiotic resistance- and virulence-related genes was obtained from the BV-BRC special gene section. Virulence and antibiotic resistance genes that were not included in BV-BRC or with specific mutations that confer antibiotic resistance (such as *ccrA, copB, grlA, gyrA,* and *vanAR*) or were involved in biofilm production (*icaABCDR, finbA, fnbB, cflA, ebpS*, or *cna*) were annotated by direct search using BLAST + 2.13.0, with known reference genes (31, 36–44).

## Pangenome modeling and phylogenetic analysis

To construct a phylogeny of the *S. aureus* strains examined in this study, we identified a set of core proteins shared by 110 *S. aureus* genomes, including 9 reference genomes (Table S3) and 101 genomes from our study, using the Bacterial Pangenome Analyses tool (BPGA v1.3) (45) (Table S3). A phylogenetic tree of *S. aureus* was obtained by aligning 2110 core proteins using MUSCLE v5.1 (46). The maximum-likelihood method was used to infer the tree using the software package IQ-TREE v2.1.2, with default parameters, 1,000 bootstrap replicates, and the JTT + F + R10 model (39). The resulting phylogenetic trees were drawn and edited using the Interactive Tree of Life (iTOL v6.6) program (47). Functional classification was performed for the core and accessory/unique proteins from the 110 *S. aureus* genomes using the NCBI batch Web CD-Search tool (https://www.ncbi.nlm.nih.gov/Structure/bwrpsb/bwrpsb.cgi) against the Clusters of Orthologous Groups (COG) database within the Conserved Domain Database (CDD) (48).

## Statistical analysis

The relative proportions of COG functional categories within the core and accessory components of the pangenome were visualized using the ggplot2 package (v4.0.1) in R (v4.5.1). Differences in the distribution of COG categories between the core and accessory genomes were assessed using two-proportion Z-tests for each category. The resulting *P*-values were adjusted for multiple tests using the Benjamini–Hochberg false discovery rate (FDR) correction. All analyses were performed using R (v4.5.1) (see Fig. S3 at https://github.com/aguayo-alan/Martinez-Flores_et_al_Microbiology_Spectrum_2026).

Differences in the number of virulence and antibiotic resistance genes between MSSA and MRSA isolates were evaluated using generalized linear mixed models (GLMMs) implemented in the glmmTMB package (v1.1.13) with a Poisson error (49). The clonal complex was included as a random effect to account for phylogenetic non-independence among isolates. MSSA was the reference category. Statistical significance was set at $\alpha = 0.05$. Model results are reported as regression coefficients ($\beta$) with standard deviation (SE).

## MLST typing and CC grouping were performed

For most isolates, the Sequence Type (ST) was obtained using the Genomic Annotation Service at BV-BRC, which assigns types by comparing them with previously reported ST alleles in PubMLST (https://pubmlst.org/) (44). If the allele was not included in PubMLST, the service of the Center for Genomic Epidemiology (https://cge.food.dtu.dk/services/MLST/) was used with default parameters and input dna.fasta files from BV-BRC to deduce the allelic profile and ST of the strain. To cluster closely related STs into CCs, we used the eBURST algorithm available at PubMLST (https://pubmlst.org/organisms/staphylococcus-aureus).

## *spa* typing

Typing of *the S. aureus* membrane protein A-coding gene (*spa*) was performed by uploading the *spa* dna.fasta files of each isolate to the spaTyper 1.0 web server (https://cge.food.dtu.dk/services/spaTyper/). The order of the specific repeats determines the *spa*

type, and a numerical code is assigned according to the nomenclature provided on the Ridom website (https://spaserver.ridom.de/).

## SCC*mec* classification

The SCC*mec* elements present in the isolates were identified using the SCCmecFinder 1.2 server. The assembled DNA sequences were uploaded onto the server with the default parameters: a 90% (ID) threshold, a minimum length of 60%, and reference as database selected. SCC*mec* cassette typing was performed according to the classification system registered in the SCCmecFinder database (https://cge.food.dtu.dk/services/SCCmecFinder-1.2/).

## RESULTS

### Clinical origin and sampling of *Staphylococcus aureus* isolates

To investigate the genomic diversity of *S. aureus* in hospital-associated infections, we analyzed 101 isolates corresponding to 2006–2019 from three tertiary care hospitals in Mexico: INPer, INC, and IDF, as described in the Materials and Methods section. Oxacillin-susceptible isolates constituted the majority of isolates in the collection compared to the resistant ones, independent of the sample source and year (Fig. 1; also see Table S1 at https://github.com/aguayo-alan/Martinez-Flores_et_al_Microbiology_Spectrum_2026). In 2013, the number of oxacillin-resistant isolates from INPER increased markedly (Fig. 1A; see below). In addition, 271 of the 286 isolates were screened for the presence of mecA by PCR. Only isolates that were both oxacillin-resistant and *mecA*-positive ($n = 50$) were classified as MRSA. Isolates displaying either oxacillin resistance (18/68) or *mecA* positivity alone were not considered to be MRSA. To perform whole-genome sequencing, a set of 101 *S. aureus* isolates was obtained from the full collection of 286 isolates (see Materials and Methods).

### Genome and pangenome structures

Whole-genome sequencing of 101 *S. aureus* isolates was performed using Illumina technology. The resulting *de novo* assemblies had a median of 44 contigs and an N50 exceeding 285 kb, yielding draft genomes approximately 2.8 Mb in size (see Fig. S1 at https://github.com/aguayo-alan/Martinez-Flores_et_al_Microbiology_Spectrum_2026). Average Nucleotide Identity by MUMmer (ANIm) comparisons showed 97%–99% identity across all genome pairs, with a GC content of 33%, consistent with known values for this species and confirming the taxonomic identity of the isolates as *S. aureus* (see Fig. S2 at https://github.com/aguayo-alan/Martinez-Flores_et_al_Microbiology_Spectrum_2026).

To identify the functional differences between the core and accessory components, we modeled the pangenome using the BPGA v1.3 program (45). The core genome component was predicted to consist of 2,110 gene-coding proteins (see Fig. S4 at https://github.com/aguayo-alan/Martinez-Flores_et_al_Microbiology_Spectrum_2026) for the set of 101 *S. aureus* isolates in this study, and nine reference genomes from different sources were obtained from GenBank (see Tables S3 and S4 at https://github.com/aguayo-alan/Martinez-Flores_et_al_Microbiology_Spectrum_2026). Furthermore, the accessory component consisted of approximately 4,671 protein families. Analysis of the accessory genome revealed an overrepresentation of genes belonging to COG categories X (mobilome: prophages and transposons), V (defense mechanisms), K (transcription), and L (replication, recombination, and repair), with respect to the core genome COGs, likely reflecting key aspects of the adaptive potential of *S. aureus* (see Fig. S3 at https://github.com/aguayo-alan/Martinez-Flores_et_al_Microbiology_Spectrum_2026).

### Phylogenetic relatedness among MRSA and MSSA isolates

To assess the genomic variability and evolutionary relationships among the MRSA and MSA isolates, we constructed a maximum likelihood phylogenetic tree based on a shared

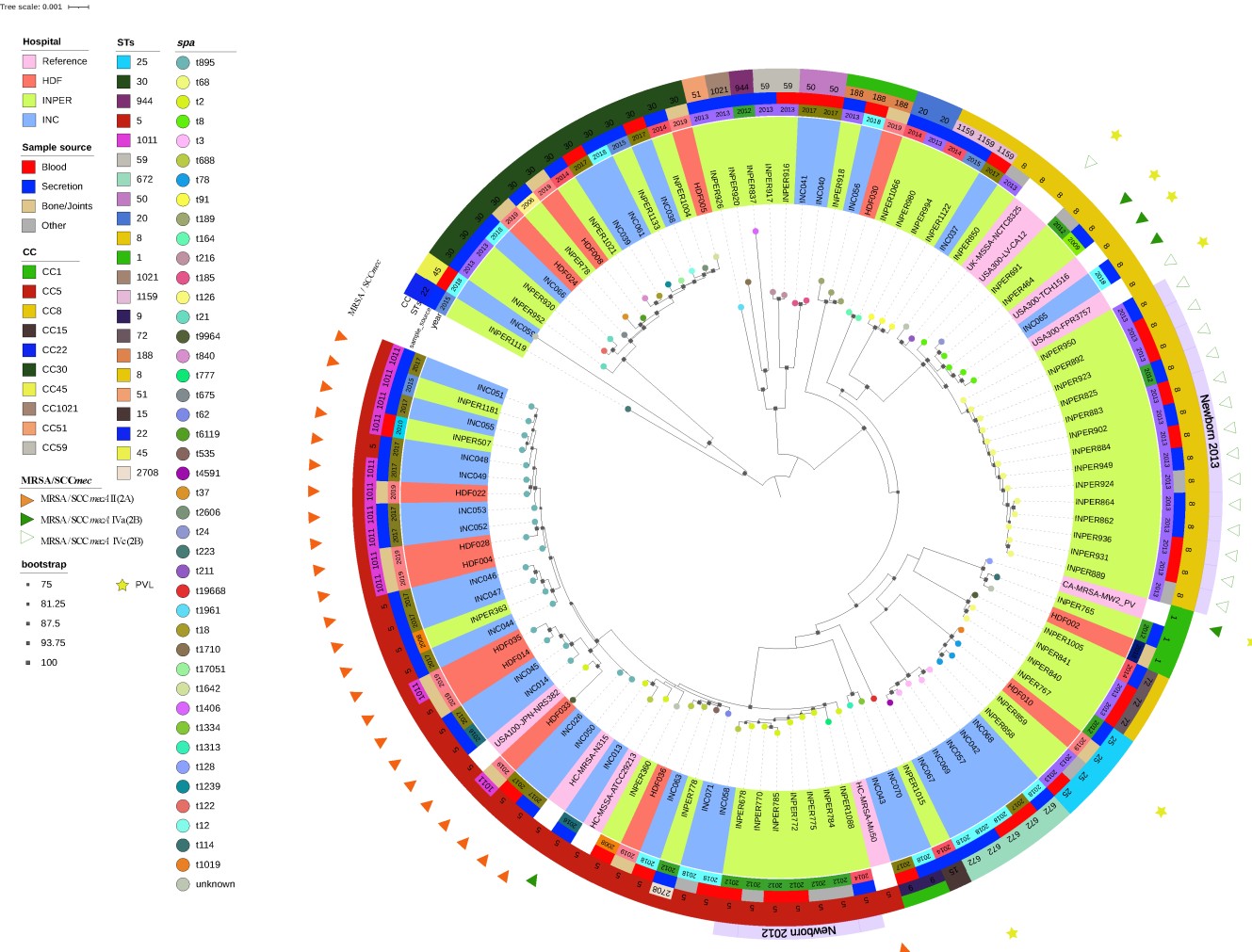

**FIG 2** Phylogenetic relationships among *S. aureus* isolates from three hospitals in Mexico. Maximum-likelihood phylogenetic tree of *S. aureus* genomes with 2,110 predicted core proteins from 101 genomes from this study and 9 reference genomes. Bootstrap scores greater than 75% are indicated by a black square at the bottom of the branches. At the tips of the branches, the *spa* types are represented by colored dots. From the innermost to the outermost rings: 1, *S. aureus* isolates coded in color according to their hospital of origin: Instituto Nacional de Perinatología (INPer) green, Instituto Nacional de Cardiología (INC) blue, and Hospital Darío Fernández (HDF) red, and reference *S. aureus* strains in pink; 2, year of isolation; 3, infection source; 4, STs and CCs as indicated by the color code in the inset; 5, clonal transmission events (newborns 2012 and 2013) in lilac ring segments; 6, MRSA isolates are marked by the presence of SCC*mec* types in red, green, red, and white triangles. The absence of the triangle indicates MSSA isolates; 7, isolates with PVL, indicated by yellow stars. All color codes are displayed in the inset.

core of 2,110 predicted proteins from the pangenome, as identified by the BPGA (Fig. 2). The resulting phylogeny revealed a diverse assemblage of closely related and highly divergent isolates organized into several major clades. Both MRSA and MSSA isolates encompassed well-separated clades associated with 21 distinct multilocus sequence types (MLSTs or STs) (Fig. 2).

The most prevalent STs were ST5, ST8, ST30, and ST1011, which clustered into Clonal Complexes CC5, CC8, and CC30, accounting for 72% of the isolates. The remaining 17 STs, comprising 28% of the isolates, belonged to 14 different CCs isolated in different years and hospitals (see Fig. S5 at https://github.com/aguayo-alan/Martinez-Flores_et_al_Microbiology_Spectrum_2026). Thus, CCs were associated with the clades defined in the phylogeny, consistent with the expected monophyletic origin of the CCs. Less-represented CCs were distributed among more divergent clades (Fig. 2; Fig. S5). This

pattern suggests that the most prevalent CCs have successfully expanded within the three hospital settings, indicating potential targets for infection control efforts.

## Distribution of MRSA isolates

To determine the association between genomic variation and methicillin resistance among hospital *S. aureus* isolates, we analyzed 39 MRSA isolates clustered into two major clonal complexes: CC5 (23/39) and CC8 (16/39). They carried SCC*mec* types II and IV, which have been associated with methicillin resistance (Fig. 2). Most CC5-MRSA isolates belonged to the Institute of Cardiology (INC) collected between 2016 and 2018 from the orthopedic hospital (HDF), and only three strains were isolated from INPer. The INPer isolates recovered in 2008 (ST5) and 2010–2015 (ST1011) were obtained from blood and wound secretions and were intermixed within the CC5-MRSA clade. Across all isolates, the hospital of origin and infection class were not associated with particular STs or CCs, suggesting that no specific adaptations were associated with the observed genomic diversity.

In contrast, a clade of highly related CC8-MRSA isolates was recovered from the blood and respiratory secretions of newborns at INPer in 2013 (Fig. 2). These isolates carried the SCC*mec* IVc element, a variant related to USA300 clones that typically harbors ACME or COMER elements and the Panton-Valentine leukocidin (PVL) toxin (50, 51). However, neither ACME/COMER nor PVL genes were detected among CC8-MRSA isolates from newborns at INPer in 2013 (see Fig. S6 at https://github.com/aguayo-alan/Martinez-Flores_et_al_Microbiology_Spectrum_2026). This lineage likely reflects either a localized transmission event within the hospital ward or the circulation of a prevalent lineage in this particularly vulnerable patient group (Fig. 2, "Newborn-2013" segment).

## MSSA isolate genome diversity

In contrast to MRSA isolates, which were limited to two clonal complexes (CCs) and three sequence types (STs), MSSA isolates exhibited greater genetic diversity and were widely distributed across the three hospitals. The most prevalent CCs among MSSA isolates were CC30 and CC5, each representing 21% of the total, whereas CC1, CC8, and other 11 sporadic CCs and STs were less frequently observed (Fig. 2; also see Fig. S5 at https://github.com/aguayo-alan/Martinez-Flores_et_al_Microbiology_Spectrum_2026).

MSSA isolates classified within CC5 but lacking the SCC*mec* element belonged to two clades closely related to the CC5-MRSA lineage (Fig. 2). One of these CC5-MSSA clades comprised isolates collected in 2012 from newborn infections at INPer hospital (Fig. 2, "Newborn-2012" segment), likely reflecting a localized transmission event within the neonatal ward, similar to what was observed in 2013, although involving CC8-MRSA isolates (see above).

Furthermore, *spa* typing revealed a high diversity of *spa* types in MSSA (40 *spa* types) compared to MRSA isolates (5 *spa* types) (Fig. 2; also see Fig. S7 at https://github.com/aguayo-alan/Martinez-Flores_et_al_Microbiology_Spectrum_2026). MSSA isolates belonging to CC30 were recovered from different sources of infection across all three hospitals (INPer, INC, and HDF). Twelve distinct *spa* types were identified among the 13 isolates in this complex, highlighting both the diversity and shared evolutionary background of these strains, which clustered into a single clade (Fig. 2).

## High prevalence of antibiotic resistance genes in MRSA isolates

To assess the differences in antibiotic resistance gene content between MRSA and MSSA isolates, we annotated the resistance genes using data from the CARD, NDARO, and BV-BRC AMR databases. Additionally, we performed targeted searches for mutations in housekeeping genes that encode antibiotic targets, such as *gyrA* and *rpoB*. Except for five MSSA isolates among the 101 *S. aureus* strains, all isolates harbored the *blaZ* gene, which encodes a beta-lactamase and confers resistance to penicillin. MRSA genomes tended to encode a higher number of antibiotic resistance genes than MSSA genomes, with mean values of 4.8 and 1.6 genes per genome, respectively. This difference was

statistically supported by the generalized linear mixed model analysis (*P* = 0.001) (Fig. 3; also see Table S6 at https://github.com/aguayo-alan/Martinez-Flores_et_al_Microbiology_Spectrum_2026).

As expected, methicillin resistance was associated with the presence of the SCC*mec* cassette in all MRSA strains, specifically CC5 and CC8. These isolates also exhibited ciprofloxacin resistance. Resistance to ciprofloxacin was associated with a *gyrA* S84L single nonsynonymous change, which was consistently identified in all MRSA strains. This mutation has been widely reported in clinical isolates and is known to reduce susceptibility to fluoroquinolones. Additionally, MRSA strains from CC5 harbored additional resistance genes not found in CC8 strains, including those associated with resistance to clindamycin and spectinomycin, whereas CC8 MRSA isolates showed resistance to tetracyclines (Fig. 4).

In contrast, MSSA isolates harbored fewer resistance genes, which were more variably distributed across different sequence types and clonal complexes (Fig. 4). Although the gene profiles varied, resistance genes for clindamycin and spectinomycin were among the most frequently detected in MSSA isolates. INPer isolates from the CC5-MSSA clade collected in 2012 carried genes conferring resistance to gentamicin and chloramphenicol, but no other resistance genes beyond *blaZ*. Isolates resistant to gentamicin and chloramphenicol almost all belonged to CC5. Resistance genes for mupirocin, macrolides (*mph*), and rifampicin were rarely found in the isolates (Fig. 4).

Notably, no resistance genes associated with trimethoprim-sulfamethoxazole, kanamycin, or vancomycin were detected in either the MRSA or MSSA isolates, suggesting a low prevalence or absence of genetic determinants for resistance to these drugs in the sampled population.

## Virulence determinants of MSSA and MRSA strains

To investigate the differences in virulence gene content between MRSA and MSSA isolates, we analyzed 115 virulence-associated genes using the VICTORs and BV-BRC

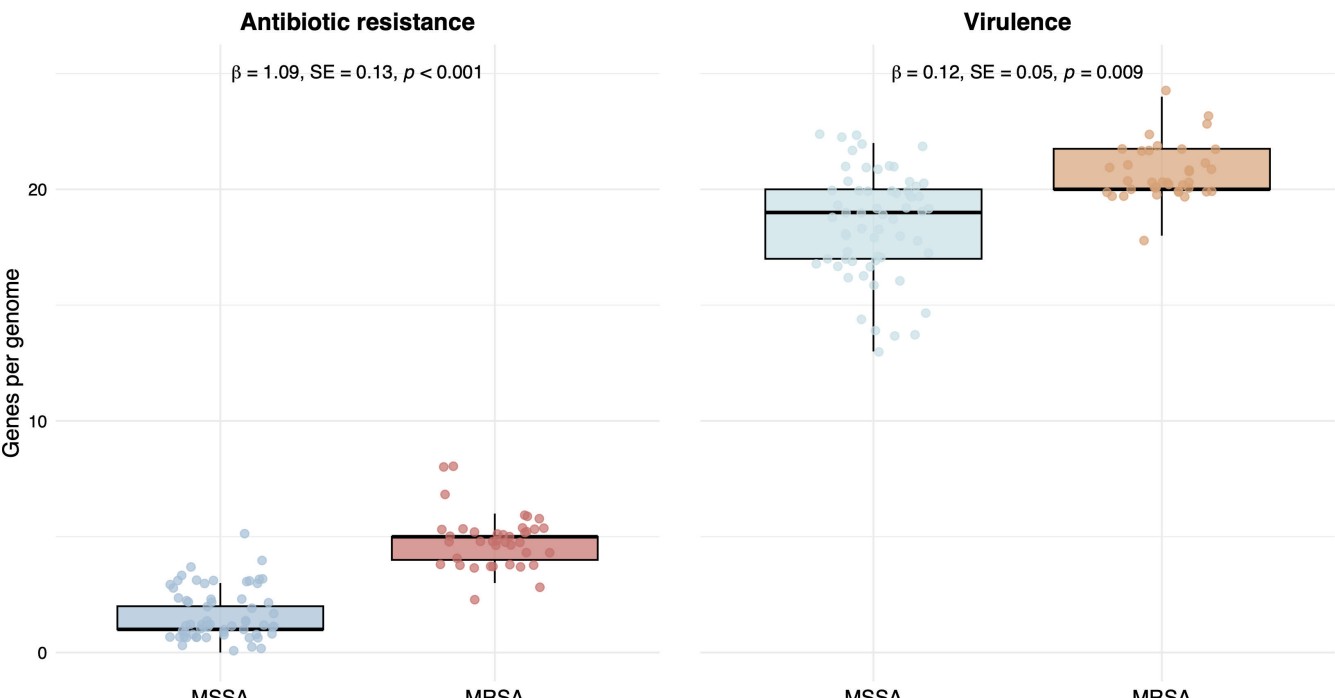

**FIG 3** Distribution of antibiotic resistance and virulence genes in MSSA and MRSA isolates. *X*-axis, MSSA (*n* = 62) and MRSA (*n* = 39). *Y*-axis: number of genes per genome in the box plot diagram. The statistics are shown on the top: *β* = regression coefficient; SE = Standard deviation; *P* = *P*-value. The details of the analysis are presented in Table S6 at https://github.com/aguayo-alan/Martinez-Flores_et_al_Microbiology_Spectrum_2026.

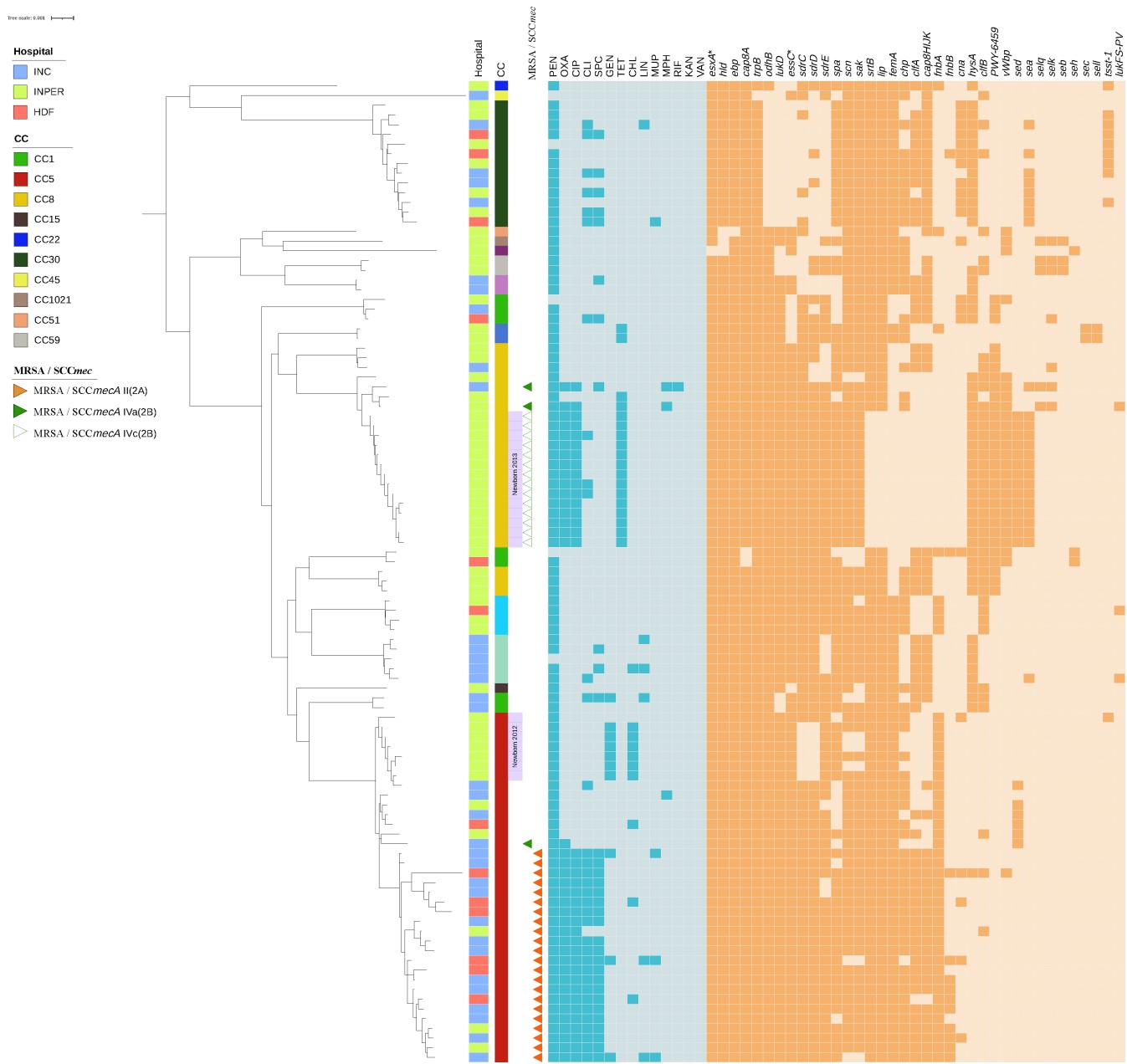

**FIG 4** Genomic profile of antibiotic resistance and virulence genes in 101 *Staphylococcus aureus* isolates from Mexico. The phylogeny of the 101 *S. aureus* isolates is shown on the left side of the figure. The inset indicates the color codes for hospital, clonal complexes (CC), and MRSA. Lilac segments indicate transmission events for Newborn 2013 and Newborn 2012. The presence of genes predicted to encode antibiotic resistance in 101 *S. aureus* isolates, as described in the Materials and Methods section, is illustrated by dark blue squares. Light blue squares indicate absence. Abbreviations for the antibiotics (upper row) are as follows: PEN, penicillin; OXA, oxacillin; CIP, ciprofloxacin; CLI, clindamycin; TET, tetracycline; SPC, spectinomycin; GEN, gentamicin; CHL, chloramphenicol; LIN, lincomycin; MUP, mupirocin; MPH (macrolide phosphotransferase: clarithromycin, erythromycin); RIF, rifampicin; KAN, kanamycin; VAN, vancomycin. The presence of virulence-associated genes in 101 *S. aureus* isolates (see Materials and Methods) is denoted by deep orange squares, and their absence is indicated by light orange squares. The names and functional annotations of the virulence genes included in this analysis are provided in Fig. S8 at https://github.com/aguayo-alan/Martinez-Flores_et_al_Microbiology_Spectrum_2026.

annotation platforms (see the Materials and Methods section). A broad array of virulence-related genes was detected, including core metabolic enzymes (e.g., *aroA*, *asd*, *thyA*, *femA*, *fbp*, and *adhB*), proteases (*clpA* and *clpX*), and transcriptional regulators (*ccpA*

and *agr*). These genes likely form a conserved virulence repertoire shared across all *S. aureus* isolates examined.

In addition, several well-known, unevenly distributed virulence determinants were identified. The median number of variable virulence genes per MRSA and MSSA isolate was 20.8 and 18.5, respectively. This difference was statistically supported by the generalized linear mixed model analysis (*P* = 0.001) (Fig. 3; also see Table S6 at https://github.com/aguayo-alan/Martinez-Flores_et_al_Microbiology_Spectrum_2026).

Within the same CC, as for CC5, MRSA strains had an average of 21 virulence genes compared to 18 in MSSA strains of the same complex; MRSA CC8 had 21 virulence genes, while MSSA CC8 had 20. Among these are the coagulase gene (*coa*), immunoglobulin-binding protein gene (*spa*), adhesion-related genes (*ebp*), capsular biosynthesis genes (*cap8A*), and components of the Type VII secretion system (*ess*) (Fig. 4; also see Table S5 at https://github.com/aguayo-alan/Martinez-Flores_et_al_Microbiology_Spectrum_2026).

Biofilm formation, which relies on the *icaABCD* operon responsible for producing polysaccharide intercellular adhesin, was detected in all isolates. However, only 77% displayed strong or moderate biofilm formation in a semiquantitative assay, whereas 23% were weakly or nonadherent (see Table S7 at https://github.com/aguayo-alan/Martinez-Flores_et_al_Microbiology_Spectrum_2026). Non-adherent strains were primarily found in CC5 (*spa* t895, SCC*mec* II [2A]), whereas CC8 strains (*spa* t068, SCC*mec* IVc [2 B]) consistently produced strong biofilms.

Other regulatory factors, such as *agrD*, may also contribute to variations in virulence. The *agrD* gene, which is part of the quorum-sensing system that modulates multiple virulence traits, was present in two *agr* types and three subtypes across the collection, potentially influencing strain-specific phenotypes, such as biofilm production (Fig. 4; also see Fig. S8 at https://github.com/aguayo-alan/Martinez-Flores_et_al_Microbiology_Spectrum_2026). Fibronectin-binding proteins *fnbA* and *fnbB*, which mediate host cell adhesion, were present in CC5 isolates but absent in CC8 and CC30 strains. The presence of *coa* corresponded to phenotypic coagulase activity (Fig. 4; also see Table S5 at https://github.com/aguayo-alan/Martinez-Flores_et_al_Microbiology_Spectrum_2026). Toxin genes of phage origin, including *lukPV*, *sea*, and *sek*, were sporadically distributed among the strains. The Panton-Valentine leukocidin gene (*lukPV*) was detected in only three isolates, which is lower than the frequencies reported in previous studies. In contrast, the *sea* gene was predominantly associated with CC8 strains.

These observations underscore the influence of clonal lineages on the virulence gene profiles of *S. aureus*, with conserved core elements complemented by variably distributed accessory genes. Such genetic variations likely contribute to the diverse pathogenic behaviors observed in clinical *S. aureus* infections.

## DISCUSSION

Our findings show that methicillin-resistant (MRSA) and methicillin-susceptible (MSSA) *Staphylococcus aureus* isolates differ substantially in their antibiotic resistance profiles yet display only modest differences in virulence gene content. While this trend has been documented in other studies, our study uniquely compared MRSA and MSSA isolates collected over more than a decade from three tertiary care hospitals in Mexico City. We found that MRSA isolates were limited to two clonal complexes (CC5 and CC8), whereas MSSA strains exhibited broader genetic diversity. Overall, no strong or consistent association was observed between specific strain types and their infection sources. Both MRSA and MSSA isolates were recovered from a broad range of infections, including bloodstream, skin and soft tissue, and other clinical sources, suggesting that multiple lineages contribute to diverse types of infection.

The *S. aureus* isolates analyzed in this study were collected between 2006 and 2019 across three tertiary-care hospitals in Mexico City and archived in a collection at the Faculty of Medicine, UNAM (Mexico City). Oxacillin-resistant isolates were recovered at relatively low frequencies throughout the study period, with occasional increases

that likely reflect the local expansion of specific resistant clones rather than sustained changes in incidence.

Genomic analyses were conducted on a set of 101 isolates from the full collection of 286 isolates to frame antibiotic diversity and virulence across MSSA and MRSA data sets. However, while the overall MRSA/MSSA ratio was preserved in the set of 101 isolates from all hospitals and study years were included, the set was not fully representative at the level of individual infection sites, with more frequent clinical sources such as blood and tissue secretions being proportionally overrepresented. The use of retrospectively collected isolates, rather than a prospective sequential sampling design, represents a limitation of this study, as it may introduce unrecognized sampling biases that could affect the observed prevalence patterns. However, the subsampling strategy was designed to minimize the overrepresentation of individual clones or narrowly defined clinical sources rather than to enable epidemiological inference. Consequently, the genomic data set does not reflect the incidence or prevalence of *S. aureus* infections in these hospitals. Instead, it provides a framework for comparative genomic analyses of MRSA and MSSA strains and captures the major genomic features and evolutionary patterns circulating in clinical settings.

An important aspect of this study is the composition of the *S. aureus* collection, which included a high proportion of pediatric isolates from the National Institute of Perinatology (INPer). In this hospital, oxacillin-resistant isolates were relatively infrequent (38/189; 20%), resulting in a predominance of MSSA in pediatric cases. In contrast, isolates from the Hospital de Traumatology and Orthopedics (HDF) and the National Institute of Cardiology (INC) were predominantly derived from adult patients and showed a higher prevalence of MRSA. Consequently, the overall MSSA/MRSA proportions reflect a composite of distinct patient populations, which likely explains the differences from studies focused primarily on adult care settings.

Methicillin-resistant strains have been reported in Mexican hospitals since 1989 (52), with CC5-MRSA (New York–Japan clone) consistently identified as the most prevalent lineage associated with hospital-acquired infections (53, 54). Although CC8-MRSA strains have also been detected, they are less frequent. Arias et al. proposed that the North American variant USA300 likely circulates in Mexico (55). In this study, we documented the clonal dissemination of CC8-MRSA among newborns at INPer Hospital in 2013. The infections occurred over several months and were likely caused by a single clone. Phylogenomic analyses confirmed the clonality of these isolates and their relationship to USA300 (North American Epidemic) and USA300-LV (Latin American variant, LV), also known as USA300-SAE (South American variant, SAE) (50, 56). These isolates lacked the COMER cassette and *lukF* and *lukS* genes (encoding PVL), which are typically found in strains from Colombia, Venezuela, and Ecuador (51, 56). We hypothesized that these CC8-MRSA strains might have acquired the SCC*mec* IVc cassette without the COMER element. In contrast, we also identified two CC8-MRSA strains resembling the USA300-NAE clone, carrying SCC*mec* IVa along with the ACME element and PVL genes (50), a variant common in North America and reported in Mexico (51, 55, 57). These two isolates (INPER464 and INC065), recovered from skin abscesses in adult patients, displayed genomic features characteristic of community-associated MRSA (CA-MRSA). As a substantial number of MSSA isolates were obtained from tissue secretions (including abscesses), it is plausible that they also originated from community-acquired infections.

The prevalence of MRSA infections varies widely across countries and healthcare settings, often contributing to high morbidity and mortality due to multidrug resistance (58, 59). Although MSSA strains are responsible for most *S. aureus* infections in hospitals, they often receive less attention because of their broad antibiotic susceptibility (60). Nonetheless, we recovered seven CC5-MSSA isolates from newborns in INPer in 2012, representing a likely case of clonal transmission within a single ward over the course of the year. Similar to CC8-MRSA, the CC5-MSSA clone appeared to undergo rapid changes through the acquisition and loss of mobile genetic elements, including phages and accessory genes (see Table S8 at https://github.com/aguayo-alan/

Martinez-Flores_et_al_Microbiology_Spectrum_2026). Such genetic variations, fueled by horizontal gene transfer, have been well documented in hospital pathogens (61).

Our findings highlight the interplay between the core and accessory genomes in shaping the diversity and pathogenic potential of *S. aureus* (62, 63). The accessory genome was enriched for prophage-related genes, transposases, and regulators, consistent with their roles in driving phenotypic heterogeneity and adaptation in hospital settings. This genomic plasticity was evident in the resistance and virulence profiles: while MRSA isolates carried a higher burden of antibiotic resistance determinants, the virulence gene patterns were more variable and dispersed. An unexpected feature was the presence of the *cap8* locus in CC5 genomes, in contrast to previous reports of *cap5* in CC5-MRSA from Mexico, suggesting local genomic diversification (57). Strong associations between *spa* types and MRSA clonal complexes further reflect the reduced diversity of MRSA compared to MSSA, supporting the idea of lineage-specific evolutionary constraints rather than sampling bias (64). At the phenotypic level, biofilm formation was not strictly correlated with the *ica* operon, emphasizing the multifactorial nature of this trait and the contribution of additional adhesion and invasion genes (*clfA, cna, fnbA,* and *ebpS*). Together, these results underscore how horizontal gene transfer, selective pressures, and lineage-specific constraints converge to generate the observed variability in clinical *S. aureus* isolates.

From a clinical standpoint, the rapid and accurate identification of virulent *S. aureus* strains is critical for timely treatment. Molecular typing methods targeting conserved and variable genes have largely supplanted traditional microbiological techniques (65–68). The detection of *mecA* is a robust predictor of methicillin resistance, consistent with our findings that resistance to methicillin coincides with the presence of the *mecA* gene (59). However, the presence of *ica* genes did not correlate reliably with biofilm formation, confirming that adherence depends on a broader set of factors (69–71).

Developing genomic reference frameworks for *S. aureus* in local hospital settings would enhance outbreak surveillance and infection control efforts. This retrospective genomic analysis demonstrates that, beyond the binary classification of MRSA and MSSA, detailed genomic profiling is essential for understanding clonal persistence, adaptation, and spread in healthcare environments. Future studies should explore how genomic variation shapes the clinical and epidemiological dynamics of *S. aureus* in Mexican hospitals.

## ACKNOWLEDGMENTS

The authors recognize the Instituto Nacional de Perinatología (INPer), Instituto Nacional de Cardiología "Ignacio Chávez" (INC) and Regional Hospital (Hospital Regional "Darío Fernández" (HDF), Miguel Ángel Cortés Mora, Orthopedic Service Head, and Joaquín González Monroy, Microbiology Laboratory Coordinator, for providing the *S. aureus* strains used in this work. They thank Shirley Alquicira and José Espíritu for their computational assistance in this study as well as Santiago Castillo for critically reading the manuscript.

The genome sequencing work was supported by the Programa de Apoyo a Proyectos de Investigación e Innovación Tecnológica (PAPIIT-UNAM) IN214019 to R.C.-C. and V.G., and complemented by the research budget of Centro de Ciencias Genómicas UNAM to V.G., and by the Departamento de Salud Pública de la Facultad de Medicina, UNAM (Proyecto No. 47/2024 DI) to R.C-C. A.A-G. is a doctoral student from the Programa de Doctorado en Ciencias Biomédicas, UNAM, with a CONAHCYT fellowship (1023079).

Irma Martínez-Flores: Formal analysis, Methodology, Investigation, Writing—original draft, Writing—review and editing. Patricia Bustos: Methodology, Investigation, Software, Visualization, Writing—review amd editing. Rosa I. Santamaría: Methodology, Software, Writing—review, and editing of the manuscript. Alan Aguayo-González: Methodology, Visualization. Eugenia Silva-Herzog: Conceptualization, Writing—review and editing the manuscript. Xavier Soberón: Conceptualization, Writing, reviewing, and editing of the manuscript. Víctor González: Funding, Investigation, Conceptualization, Formal Analysis,

Writing—original draft, writing, review, and editing. Roberto Cabrera-Contreras: Funding, Conceptualization, Resources. Writing, review, and editing.

PaperPal (Cactus Communications Services Pte, LTD) was used to correct English grammar. The authors are fully responsible for the originality and content of the manuscript.

## AUTHOR AFFILIATIONS

[1]Centro de Ciencias Genómicas, Universidad Nacional Autónoma de México, Cuernavaca, Morelos, México

[2]Unidad de Vinculación Científica, Facultad de Medicina UNAM en Instituto Nacional de Medicina Genómica (INMEGEN), Ciudad de México, México

[3]Instituto de Biotecnología, Universidad Nacional Autónoma de México, Cuernavaca, Morelos, México

[4]Laboratorio de Patogenicidad Bacteriana, Departamento de Salud Pública, Facultad de Medicina, Universidad Nacional Autónoma de México, Ciudad de México, México

## AUTHOR ORCIDs

Irma Martínez-Flores  http://orcid.org/0000-0001-9350-8978
Patricia Bustos  http://orcid.org/0000-0002-6924-8747
Rosa I. Santamaría  http://orcid.org/0000-0001-6970-7336
Alan Aguayo-González  http://orcid.org/0009-0000-8925-2164
Eugenia Silva-Herzog  http://orcid.org/0000-0001-5620-8722
Xavier Soberón  http://orcid.org/0000-0002-4498-5628
Víctor González  http://orcid.org/0000-0003-4082-0022
Roberto Cabrera-Contreras  http://orcid.org/0000-0001-6734-1967

## FUNDING

| Funder | Grant(s) | Author(s) |
|---|---|---|
| Programa de Apoyos a Proyectos de Investigación e Innovación Tecnológica (PAPIIT-UNAM) | IN214019 | Víctor González |
| | | Roberto Cabrera-Contreras |

## AUTHOR CONTRIBUTIONS

Irma Martínez-Flores, Data curation, Formal analysis, Investigation, Methodology, Resources, Software, Validation | Patricia Bustos, Data curation, Formal analysis, Methodology, Resources, Software, Validation, Visualization | Rosa I. Santamaría, Data curation, Methodology, Resources, Software | Alan Aguayo-González, Methodology, Resources, Validation, Visualization | Eugenia Silva-Herzog, Conceptualization, Formal analysis, Investigation, Resources, Visualization, Writing – original draft | Xavier Soberón, Conceptualization, Data curation, Formal analysis, Investigation, Writing – original draft | Víctor González, Conceptualization, Formal analysis, Funding acquisition, Investigation, Project administration, Supervision, Validation, Writing – original draft, Writing – review and editing.

## DATA AVAILABILITY

The genome sequences of the *S. aureus* strains were uploaded to GenBank, and accession numbers are provided in Table S4 at https://github.com/aguayo-alan/Martinez-Flores_et_al_Microbiology_Spectrum_2026.

## ETHICS APPROVAL

This study was approved by the Ethics Committee of the Faculty of Medicine of the Universidad Nacional Autónoma de México.

## ADDITIONAL FILES

The following material is available online.

### Open Peer Review

**PEER REVIEW HISTORY (review-history.pdf).** An accounting of the reviewer comments and feedback.

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
