## [Reviewer comments · Microbiology Spectrum]

Microbiology Spectrum

Comparative Genomics of Methicillin-Resistant and - Susceptible *Staphylococcus aureus* from México City Hospitals

Irma Martinez-Flores, Patricia Bustos, Rosa Santamaria, Alan Aguayo González, Eugenia Silva Herzog, Xavier Soberón, Victor Gonzalez, and Roberto Cabrera-Contreras

Corresponding Author(s): Victor Gonzalez, Universidad Nacional Autonoma de Mexico - Campus Morelos

Review Timeline:

Submission Date:	October 1, 2025
Editorial Decision:	November 27, 2025
Revision Received:	January 22, 2026
Accepted:	February 5, 2026

Editor: Ayush Kumar

Reviewer(s): Disclosure of reviewer identity is with reference to reviewer comments included in decision letter(s). The following individuals involved in review of your submission have agreed to reveal their identity: Andrew David Berti (Reviewer #1)

Transaction Report:

DOI: <https://doi.org/10.1128/spectrum.03140-25>

Re: Spectrum03140-25 (**Comparative Genomics of Methicillin-Resistant and -Susceptible *Staphylococcus aureus* from México City Hospitals**)

Dear Dr. Victor Gonzalez:

Thank you for the privilege of reviewing your work. Your manuscript has been reviewed by two experts in the field and they both express their enthusiasm for your work. However, they have also suggested a number of minor modifications and I would like you to address those before I can consider your manuscript for publication. Below you will find instructions from the Spectrum editorial office, and the reviewer comments.

Revision Guidelines

Sincerely,
Ayush Kumar
Editor
Microbiology Spectrum

Reviewer #1 (Comments for the Author):

I read with interest the contribution by Martínez-Flores et al. (msystems.224389). The authors perform a series of genomic and phylogenetic analyses of MSSA and MRSA clinical isolates from Mexico City, 2006-2020. Descriptive studies of microbial

ecological changes over time, particularly in undersampled regions, are valuable for tracking the emergence and spread of novel genomic variants. One potential weakness of the study is the lack of any compelling new finding that would distinguish this manuscript from many similar surveillance manuscripts from other geographical areas. However, the current study has several strengths, is well-described, and distinguishes itself by both the clarity of presentation and the detailed description of both clinical cases and individual genomic analyses. Identification of clonal transmission within and between healthcare settings is also an important finding from this study. My criticisms are relatively minor and reproduced below.

Considerations:

- * While this study may not have been funded by a major federal grant or other sponsor, funds to support the sequencing effort were likely provided from some source and should be acknowledged (e.g. Centro de Ciencias Genómicas, UNAM).
- * The statement, "Oxacillin-susceptible isolates predominated..." on line 221 is confusing and could benefit from revision.
- * I appreciate that the proportion of sequenced isolates was balanced across infection types and age groups - however, specific information on how they were selected (randomly, balanced by year of isolation, balanced by hospital...) would be appropriate to ensure there was minimal selection bias. I do appreciate, however, the discussion as a limitation beginning in line 381.
- * While your discussion overall is robust and appropriate, I was surprised that there was minimal commentary regarding the different distribution of MSSA/MRSA in pediatrics compared to adults. This would help to explain why your proportions and overall findings may be distinct from other groups that sampled predominantly from adult health centers.
- * I did not find any mention on whether this collection contained all sequential isolates or if there was any potential for selection bias during sample collection. I would assume based on the hospital location that you would recover significantly more than 200-300 clinical isolates over the study period. Was the collection from challenging cases or in any other way potentially biased? If so, or if unknown, I think it should be stated as a potential bias.

Reviewer #2 (Comments for the Author):

This manuscript compares the genomes of MRSA and MSSA strains isolated from Mexican hospitals. Strain types and virulence factors are analyzed.

The following comments are made:

1. Proofread the English.
2. Line 100. Italicize the scientific names. Correct throughout the text.
3. Line 106. They are usually named "Table S1 or Figure S1". Correct in all figures, tables and text
4. Table S1. Include the patient's diagnosis to determine the disease from which *S. aureus* was isolated and to allow for discussion.
5. Lines 107-109. For the strains whose genomes were not sequenced, how was it molecularly confirmed that they were *S. aureus*? Explain.
6. Line 108. Explain why you selected those 101 strains. Why you didn't sequence all the strains?
7. Why was LB medium used to isolate *S. aureus*? It is not a selective medium. Explain
8. Line 115. Gram is a proper noun; capitalize it.
9. Line 116. Resistance to what? State it.
10. Line 118. It is unclear how you determined they had MRSA strains. Normally, the oxacillin susceptibility test is performed, and CLSI defines a strain as MRSA if it has a resistance {greater than or equal to} 4 µg/mL. Clarify this.
11. Lines 123-125. State the number of CFU/mL used; this is the standard practice.
12. Lines 140-141. Consistently use "L" instead of "l" throughout the text when referring to liters, mL, and µL. Correct
13. Lines 142-143. State the RNase concentration and all solutions used.
14. Line 148. The sequencing methodology used is not specified; only the company is mentioned. State the methodology used.
15. Lines 158 and 168. Indicate the reference genomes.
16. Line 204. Indicate whether informed consent was obtained from patients for sample collection and use of results.
17. Lines 26 and 2020. What are the correct years? Correct and state.
18. Line 663. Figure 1 does not include an asterisk. Add one.
19. Line 225. The detection of the *mecA* gene does not necessarily indicate MRSA strains, as strains resistant to oxacillin at concentrations <4 µg/mL, may also possess the *mecA* gene but are not MRSA, are MSSA (as indicated by the CLSI). Therefore, this is not sufficient to make that assertion. Modify or correct.
20. Line 227. This could be expanded upon, as the reason for selecting these strains is never explained. This could be included in the Materials and Methods section.
21. Figure 1. Put the three graphs in relation to the number of strains like graph A. To be able to compare them better.
22. Figure S3. The code A is missing from the graph. Add it.
23. Lines 252-253. Figure 2 does not specify which strains are MRSA and MSSA, making it difficult to distinguish the clades mentioned. Add this information.
24. Lines 275, 290. Identify the MRSA and MSSA strains in Figure 2 for quick visualization.
25. Figure S8. This figure displays interesting data; it could be included as a regular figure, non-supplementary figure.

26. Figure 3. Indicate the MRSA and MSSA strains for quick visualization.
27. Figure 3. Italicize the gene names.
28. Line 691. Include the names of the analyzed genes, as they are not listed in the Methods section.
29. Line 350. Italicize the gene names: *spa* and *mec* from SCCmec. Correct throughout the text.
30. Line 381. Discuss the MRSA/MSSA relationship and explain the difference.
31. Lines 391-404. Discuss the possible presence of CA-MRSA strains in hospitals.
32. Lines 405-414. Discuss the types of strains found associated with the infections from which the strains were isolated.
33. Lines 416-418. Include a supplementary table listing the accessory genomes found in the strains.
34. Lines 434-436. The fact that a strain possesses the *mecA* gene does not mean it is MRSA, as it can be a strain with a MIC <4 µg/mL that possesses the gene but is MSSA.
35. In the references, use italics for scientific names.
36. Review the references to ensure they conform to the Author Guidelines and standardize them.

En este manuscrito se compararon los genomas de cepas MRSA y MSSA aisladas de hospitales mexicanos. Se analizan los tipos de cepas y factores de virulencia. Se hacen los siguientes comentarios:

1. Hacer una revisión del inglés.
2. Line 100. Poner los nombres científicos en cursivas. Corregir en todo el texto.
3. Line 106. Normalmente se pone Table S1. Corregir
4. Table S1. Poner el diagnóstico del paciente para saber a partir de que enfermedad se aislo *S. aureus* y poder discutir esto.
5. Lines 107-109. Las cepas que no se secuencio el genoma. Como comprobaron molecularmente que eran *S. aureus*? Ponerlo.
6. Line 108. Poner porque seleccionaron esas 101 cepas y no secuenciaron todas las cepas.
7. Porque usaron medio LB para aislar *S. aureus*? No es un medio selectivo
8. Line 115. Gram es un nombre propio, ponerlo en mayúscula.
9. Line 116. Resistencia a qué? Ponerlo.
10. Line 118. No queda claro como determinaron que tenían cepas MRSA. Normalmente se hace la prueba de susceptibilidad a oxacilina y el CLSI marca que una cepa es MRSA si tiene una resistencia $\geq 4\mu\text{g}/\text{mL}$. Aclararlo.
11. Lines 123-125. Poner cuantas UFC/mL se pusieron, es lo que normalmente se indica.
12. Lines 140-141. Homogenizar en todo el texto el uso de L en lugar de l cuando se habla de litros, mL, μL .
13. Lines 142-143. Poner la concentración de RNase y todas las soluciones usadas.
14. Line 148. No se indica que tipo de metodología se utilizó para la secuenciación, solo se menciona la compañía. Poner la metodología utilizada.
15. Line 158 and 168. Indicar cuales fueron los genomas de referencia.
16. Line 204. Poner si se pidió consentimiento informado a los pacientes para la toma y uso de las muestras.
17. Line 26 and 2020. Cuáles son los años correctos. Corregir y poner.
18. Line 663. En la Figure 1 no se indica ningún asterisco. Ponerlo.
19. Line 225. El que se detecte el gen *mecA* no quiere decir que sean cepas MRSA, ya que pueden tener cepas que presentan resistencia a oxacilina a una concentración $<4\mu\text{g}/\text{mL}$ que van a presentar el gen *mecA* y no son MRSA. Por lo que eso no es suficiente para poner esa aseveración. Modificar o corregir.
20. Line 227. Podrían ampliar esto, ya que nunca ponen porque seleccionaron esas cepas. Podrían ponerlo en Material y Métodos.
21. Figure 1. Poner las tres graficas en relación al número de cepas como la gráfica A. Para poder compararlas mejor.
22. Figure S3. No esta en la gráfica el código A. Ponerlo
23. Lines 252-253. En la Figura 2 no se especifica cuáles son las cepas MRSA y MSSA, para poder distinguir los clados que dicen. Ponerlo.

24. Lines 275, 290 . Poner en la Figure 2 cuales son las cepas MRSA y MSSA. Para poder visualizarlo rápidamente.
25. Figure S8. Esta figura muestra datos interesantes, podrían ponerla como Figura normal no suplementaria.
26. Figure 3. Indicar las cepas MRSA y MSSA para poder visualizarlas rápidamente.
27. Figure 3. Poner el nombre de los genes en cursivas.
28. Line 691. Poner el nombre de los genes analizados ya que no está en Métodos.
29. Line 350. Poner el nombre de los genes en cursivas: spa y mec del SCCmec.
Corregir en todo el texto.
30. Line 381. Discutir la relación MRSA/MSSA a que se debe la diferencia.
31. Lines 391-404. Discutir la posible presencia de cepas CA-MRSA en los hospitales.
32. Lines 405-414. Discutir el tipo de cepas encontradas con las infecciones de donde fueron aisladas las cepas.
33. Line 416-418. Podrían mencionar en una tabla suplementaria los genomas accesorios encontradas en las cepas.
34. Line 434-436. El que la cepa presente el gen mecA no quiere decir que es MRSA ya que puede ser una cepa que tiene un MIC $<4\mu\text{g}/\text{mL}$ que presenta el gen pero no es MRSA. Corregir
35. Poner en las referencias los nombres científicos en cursivas.
36. Revisar que las referencias estén de acuerdo a Instrucciones para autores y homgenizarlas.

Manuscript Spectrum03140-25 (revised version)

Comparative Genomics of Methicillin-Resistant and -Susceptible *Staphylococcus aureus* from México City Hospitals

Irma Martínez-Flores^{1, §}, Patricia Bustos^{1, §}, Rosa I. Santamaría¹, Alan Aguayo-González¹, Eugenia Silva-Herzog², Xavier Soberón³, Víctor González^{1, *}, Roberto Cabrera-Contreras^{4, *}

Response to reviewers.

Reviewer #1 (Comments for the Author):

I read with interest the contribution by Martínez-Flores et al. (msystems.224389). The authors perform a series of genomic and phylogenetic analyses of MSSA and MRSA clinical isolates from Mexico City, 2006-2020. Descriptive studies of microbial ecological changes over time, particularly in undersampled regions, are valuable for tracking the emergence and spread of novel genomic variants. One potential weakness of the study is the lack of any compelling new finding that would distinguish this manuscript from many similar surveillance manuscripts from other geographical areas. However, the current study has several strengths, is well-described, and distinguishes itself by both the clarity of presentation and the detailed description of both clinical cases and individual genomic analyses. Identification of clonal transmission within and between healthcare settings is also an important finding from this study. My criticisms are relatively minor and reproduced below.

Response. Thank you for your comments. While several aspects in our study are similar to those reported elsewhere, this study provides a unique long-term genomic analysis in Mexico. The manuscript presents a genomic investigation of *S. aureus* isolates collected over a 14-year period from three major hospitals in Mexico City. This longitudinal approach in a region previously characterized by fragmented reporting provides a more comprehensive understanding of the local evolution and transmission dynamics of MRSA and MSSA. Indeed, comparative genomics of 101 isolates provides a reference framework to approach both basic and clinical research problems of the biology of *Staphylococcus*.

In this work a distinct MRSA lineage in newborns is identified and characterized. A key finding is the identification of a specific clade of CC8-MRSA that caused infections among newborns in 2013. Genomic analysis revealed this lineage is related to the widespread USA300 clone but uniquely lacks the Panton-Valentine leukocidin (PVL) genes and the COMER island.

The study documents a localized transmission event of CC5-MSSA isolates among neonates in a single hospital ward in 2012. It demonstrates that the small genomic variations within this MSSA cluster were likely due to the acquisition and loss of mobile genetic elements like prophages. This suggest rapid, localized microevolution of MSSA

within a hospital setting, a dynamic that is often overlooked in surveillance efforts focused on MRSA.

However, we recognized the reviewer point on the limitations of our findings.

Considerations:

* While this study may not have been funded by a major federal grant or other sponsor, funds to support the sequencing effort were likely provided from some source and should be acknowledged (e.g. Centro de Ciencias Genómicas, UNAM).

Response: Support for genome sequencing came from UNAM Funds (Programa de Apoyo a Proyectos de Investigación e Innovación Tecnológica (PAPIIT-UNAM N-0000) complemented with funds of the Centro de Ciencias Genómicas-UNAM.

* The statement, "Oxacillin-susceptible isolates predominated..." on line 221 is confusing and could benefit from revision.

Response: Thank you for the observation. Certainly, the statement is ambiguous. We remove the statement and re-write the main message in a simple way: "Oxacillin-susceptible isolates constituted the majority of isolates in the collection compared to the resistant ones, independent of the sample source and year (l. 235-237) It matched well with the data in Fig. 1.

* I appreciate that the proportion of sequenced isolates was balanced across infection types and age groups - however, specific information on how they were selected (randomly, balanced by year of isolation, balanced by hospital...) would be appropriate to ensure there was minimal selection bias. I do appreciate, however, the discussion as a limitation beginning in line 381.

Response: To obtain the set of *S. aureus* isolates for whole-genome sequencing, we specifically preserved the temporal distribution of MRSA and MSSA isolates across the study period and ensured inclusion of isolates representing all major infection types diagnosed in the three participating hospitals (Fig. 1C). This approach was intended to minimize bias toward a single clinical source or time point. However, because isolates from certain infection types (e.g., blood and tissue secretions) were more frequent in the original collection, they are proportionally more represented in the sequenced set. We have revised the corresponding paragraph in the Material and Methods section (l. 113-122) to explicitly describe the selection criteria and clarify that the set was balanced in the representation of MRSA/MSSA, years of isolation and infection source. In addition, we expanded the Discussion (beginning at line 381; now in l. 420-433) to explicitly acknowledge that the sequenced set is not fully representative of the entire isolate collection and that this constitutes a limitation of the study.

* While your discussion overall is robust and appropriate, I was surprised that there was minimal commentary regarding the different distribution of MSSA/MRSA in pediatrics

compared to adults. This would help to explain why your proportions and overall findings may be distinct from other groups that sampled predominantly from adult health centers.

Response: Thank you for highlighting this important point. The differing distribution of MSSA and MRSA between pediatric and adult populations in our study is largely driven by the characteristics of the participating hospitals, particularly the National Institute of Perinatology (INPer), a high-risk pregnancy and neonatal referral center. INPer contributed 189 isolates to the overall collection, of which 136 (71%) were obtained from newborn patients. In this hospital, oxacillin-resistant isolates were relatively infrequent (38/189; 20%), resulting in high proportion of oxacillin sensible isolates among pediatric cases. In contrast, all isolates from the Hospital de Traumatology and Orthopedics (HDF) were derived from adult patients, and only 7 of 70 isolates from the National Institute of Cardiology (INC) originated from pediatric patients. In both HDF and INC, MRSA isolates were more prevalent and more evenly distributed across sampling years.

As a result, the overall MSSA/MRSA proportions observed in our study reflect a composite of distinct patient populations rather than a uniform hospital setting. This demographic structure helps explain why our findings differ from studies conducted primarily in adult-care hospitals, where MRSA is often more dominant. Importantly, these observations underscore that surveillance efforts should not be restricted to MRSA, particularly in pediatric and neonatal settings where MSSA remains highly prevalent and clinically relevant.

Although the proportion of MSSA and MRSA varies across hospitals due to differences in patient age and clinical focus, this structure does not compromise the comparative genomic analyses performed in this study. Our statistical framework explicitly accounts for clonal background, allowing MRSA and MSSA comparisons within lineages rather than across heterogeneous populations. Therefore, while demographic factors shape the observed prevalence of resistance phenotypes, the genomic patterns described here reflect intrinsic differences in mobilome composition and virulence gene distribution, rather than sampling artifacts.

A paragraph addressing this point was introduced in the Discussion (l. 435-443).

* I did not find any mention on whether this collection contained all sequential isolates or if there was any potential for selection bias during sample collection. I would assume based on the hospital location that you would recover significantly more than 200-300 clinical isolates over the study period. Was the collection from challenging cases or in any other way potentially biased? If so, or if unknown, I think it should be stated as a potential bias.

Response: Thank you for this important clarification. The isolates included in this study do not constitute a collection of all sequential *Staphylococcus aureus* isolates recovered at each hospital during the study period. Instead, they represent a retrospective archive of clinical isolates preserved through routine diagnostic and research activities. Because

isolate archiving was not originally designed for systematic surveillance, we cannot exclude the presence of selection bias, and this limitation is now explicitly stated in the Discussion (l. 420-433).

The total number of isolates included (286) reflects the subset that could be reliably recovered, characterized, and linked to minimal clinical metadata. Importantly, isolates were not selected based on disease severity or clinical outcome, but practical constraints may have favored the retention of isolates from specific wards, infection types, or time periods. Consistent with the clinical focus of the participating hospitals, the collection is enriched for neonatal isolates at the perinatal center and adult isolates at the cardiology and trauma hospitals.

To mitigate potential bias in downstream genomic analyses, we used a subsampling strategy that preserved MRSA/MSSA proportions across time, hospitals, and source of infection. Nevertheless, we acknowledge that the collection cannot be used to infer incidence or prevalence, and that unrecognized biases related to isolate archiving may influence observed distributions. We now explicitly state these limitations in the revised manuscript (l. 430-433)

Reviewer #2 (Comments for the Author):

This manuscript compares the genomes of MRSA and MSSA strains isolated from Mexican hospitals. Strain types and virulence factors are analyzed.

The following comments are made:

1. Proofread the English.

Response: Proofreading english and grammar was carefully examined and fully corrected.

2. Line 100. Italicize the scientific names. Correct throughout the text.

Response: Revised and corrected throughout the manuscript.

3. Line 106. They are usually named "Table S1 or Figure S1". Correct in all figures, tables and text

Response: Corrected in all the figures, tables, and text.

4. Table S1. Include the patient's diagnosis to determine the disease from which *S. aureus* was isolated and to allow for discussion.

Response: Unfortunately, we had no access to patient clinical information. The minimal information from patients was age and the sites of isolation (blood, urine, bone, and others), but the disease, treatment, and patient outcome remained anonymous.

5. Lines 107-109. For the strains whose genomes were not sequenced, how was it molecularly confirmed that they were *S. aureus*? Explain.

Response: All isolates were confirmed first as *S. aureus* using standard microbiological tests (Gram staining, mannitol fermentation, including catalase, coagulase enzymes and resistance to cefotixin and oxacillin) and molecular methods, including PCR of the *coa* and *mecA* gene. Phenotypes for oxacillin and PCR of the *mecA* gene are included in Table S1.

6. Line 108. Explain why you selected those 101 strains. Why you didn't sequence all the strains?

Response: To select the set of *S. aureus* isolates for whole-genome sequencing, we preserved the temporal distribution of MRSA and MSSA isolates across the study period and ensured inclusion of isolates representing all major infection types diagnosed in the three participating hospitals (Fig. 1C). This approach was intended to minimize bias toward a single clinical source or time point.

However, because isolates from certain infection types (e.g., blood and tissue secretions) were more frequent in the original collection, they are proportionally more represented in the sequenced set. We have revised the corresponding paragraph in the Material and Methods section (lines 113-122) to explicitly describe the selection criteria and clarify that the subset was not randomly selected. In addition, we expanded the Discussion (beginning at line 381) to explicitly acknowledge that the sequenced set is not fully representative of the entire isolate collection and that this constitutes a limitation of the study.

Due financial limitation, the budget for genome sequencing was limited to 101 *S. aureus* isolates.

7. Why was LB medium used to isolate *S. aureus*? It is not a selective medium. Explain

Response: Primary isolates were cultured in selective medium (mannitol salt medium) followed by microbiological test as Gram staining, mannitol fermentation, including catalase, coagulase enzymes and resistance to oxacillin and cefotixin. To routinely maintain *S. aureus*, they had to be cultured in a rich medium as LB. These observations were clarified in the manuscript (l. 123-127).

8. Line 115. Gram is a proper noun; capitalize it.

Response: Right, it was corrected.

9. Line 116. Resistance to what? State it.

Response: Corrected, phrase was incomplete. It was corrected as follow: “Susceptibility to oxacillin...”. (l. 130)

10. Line 118. It is unclear how you determined they had MRSA strains. Normally, the oxacillin susceptibility test is performed, and CLSI defines a strain as MRSA if it has a resistance greater than or equal to 4 µg/mL. Clarify this.

Response: All isolates were tested for oxacillin resistance and the presence of *mecA* with PCR. Oxacillin resistance was tested according to the CLSI standards as stated in Material and Methods section. In our experiments we consider that isolates with resistance above 5 µg/mL could be considered only “oxacillin-resistant” (Figure 1A). Further characterization of the isolates for the presence of *mecA* reinforces the classification of the isolates as MRSA. This preliminary screening resulted in 68/286 isolates resistant to oxacillin; 50 of them were positive for the presence of the *mecA* gene (Table S1). The rest did not show *mecA* presence. In the Figure 1A, only the isolates with the oxacillin resistant phenotype were considered. The genome of 38 out 39 MRSA isolates that were chosen for sequence presented a phenotype of oxacillin resistance and presence of the *mecA* gene.

11. Lines 123-125. State the number of CFU/mL used; this is the standard practice.

Response: In the text, this parameter was included, based on $OD_{600} = 1.8$, the inoculum used corresponded to 1.5×10^7 CFU/mL.

12. throughout the text when referring to liters, mL, and µL. Correct

Response: All these mistakes were corrected.

13. Lines 142-143. State the RNase concentration and all solutions used.

Response: The RNase concentration and all solutions used in the genome extraction procedure were now detailed in the manuscript text (l- 151-165) as follows: Genomic DNA was extracted using the GenElute Bacterial Genomic DNA Kit (Sigma-Aldrich) according to the manufacturer's protocol. Briefly, 1.5 mL of an overnight bacterial broth culture were collected and resuspended in 200 µL of lysozyme solution (lysostaphin 200 units/mL, Sigma Aldrich), and 20µL of RNase A solution (20 mg/mL) was added to obtain RNA-free genomic DNA. The cells were lysed with 200 µL lysis solution C (B8803 Sigma-Aldrich) and 20 µL Proteinase K (20 mg/mL)

14. Line 148. The sequencing methodology used is not specified; only the company is mentioned. State the methodology used.

Response: Genome sequencing was performed using BGI Tech Solution (BGI Americas Corporation, Cambridge, MA, U.S.A) with the DNA nanoball PCR free whole genome sequencing (DNBSEQ WGS). Libraries containing 350 bp inserts were sequenced, obtaining 2 Gb of 150 bases per read in a pair-end sequencing (PE) strategy.

15. Lines 158 and 168. Indicate the reference genomes.

Response: The Table S3 contains the list of *S. aureus* reference strains. The Figure S2 was corrected including the corresponding reference genomes.

16. Line 204. Indicate whether informed consent was obtained from patients for sample collection and use of results.

Response: This study was carried out following the recommendations of the ethics review committee of the Facultad de Medicina-UNAM. All *S. aureus* isolates used in this work were obtained by a donation from three tertiary care hospitals in Mexico City, the microbiology collection of the Instituto Nacional de Perinatología "Isidro Espinosa de los Reyes" (INPer), the National Institute of Cardiology (INC) and from the Hospital de Traumatología and Orthopedics (HDF). A consent form was not required. Original identification keys and clinical data concerning the isolates are maintained under the control of INPer, INC and HDF. In the present work, new strain identifiers were assigned to the INPer, INC and HDF isolates. Authors do not have access in any form to the specific clinical information of strains and patients.

17. Lines 26 and 2020. What are the correct years? Correct and state.

Response: Thank you for noting this discrepancy. The difference arises because the full archived collection comprises 286 *Staphylococcus aureus* isolates collected between 2006 and 2020, whereas the set of 101 isolates selected for whole-genome sequencing spans the period from 2006 to 2019. Figures 1A and 1B describe oxacillin resistance phenotypes and clinical origins for the entire collection (2006–2020), while Figure 1C corresponds exclusively to the sequenced set (2006–2019). We have corrected and clarified the corresponding lines in the manuscript to explicitly state these date ranges.

18. Line 663. Figure 1 does not include an asterisk. Add one.

Response: Figure 1 was modified according to the query 21 in this review. The information appears clearer in it; thus, we judge unnecessary to add the asterisk.

19. Line 225. The detection of the *mecA* gene does not necessarily indicate MRSA strains, as strains resistant to oxacillin at concentrations <4 µg/mL, may also possess the *mecA* gene but are not MRSA, are MSSA (as indicated by the CLSI). Therefore, this is not sufficient to make that assertion. Modify or correct.

Response: Thank you for this important clarification. We agree that detection of the *mecA* gene alone is not sufficient to define an isolate as MRSA, as oxacillin resistance must be interpreted according to CLSI breakpoints and phenotypic susceptibility testing. We have therefore revised the text to clarify that methicillin resistance was determined based on standard antimicrobial susceptibility testing, while *mecA* detection was used as a complementary molecular marker.

Accordingly, isolates were classified as MRSA only when oxacillin resistance met CLSI criteria, whereas isolates that were oxacillin-susceptible despite carrying *mecA* were classified as MSSA. We have corrected the relevant section of the manuscript to reflect this distinction and avoid overinterpretation based solely on *mecA* detection (l. 235-244)

20. Line 227. This could be expanded upon, as the reason for selecting these strains is never explained. This could be included in the Materials and Methods section.

Response: Thank you for inquiring about this relevant point. The set of 101 *S. aureus* isolates was obtained from the full collection of 286 isolates. The 101-subset aimed to preserve the temporal distribution of MRSA and MSSA isolates and to ensure representation across hospitals and all the major clinical infection types (Fig. 1C). We clarify in the Material and Methods section that the overall MRSA/MSSA ratio and the inclusion of isolates from all hospitals and study years were maintained; however, the set was not fully representative of specific infection sites. In particular, more frequent clinical sources, such as blood and tissue secretions, were proportionally overrepresented. Accordingly, the sequenced set was designed to support comparative genomic analyses rather than epidemiological inference.

This modification was included in the results section (l. 113-122).

21. Figure 1. Put the three graphs in relation to the number of strains like graph A. To be able to compare them better.

Response: All three graphs were modified as suggested.

22. Figure S3. The code A is missing from the graph. Add it.

Response: We revised the methodology used for COG annotation. The absence of COG category A in Figure S3 of the original manuscript was due to the use of an outdated COG database. In the revised version, we employed the most recent COG database (2024), and proteins assigned to COG category A are now included. These updates do not alter the conclusions of this section.

23. Lines 252-253. Figure 2 does not specify which strains are MRSA and MSSA, making it difficult to distinguish the clades mentioned. Add this information.

Response: Thanks. Annotations in the Figure 2 with the indication of “MRSA/SSCmec” were added. The legend was made more explicit.

24. Lines 275, 290. Identify the MRSA and MSSA strains in Figure 2 for quick visualization.

Response: The annotation “MRSA/SSCmec” was added in the Figure and the legend modified accordingly.

25. Figure S8. This figure displays interesting data; it could be included as a regular figure, non-supplementary figure.

Response: Yes, we agree. The figure was revised and incorporated in the main manuscript.

26. Figure 3. Indicate the MRSA and MSSA strains for quick visualization.

Response: Figure 3 was modified to add the indication of “MRSA/SCCmec”.

27. Figure 3. Italicize the gene names.

Response: Names were italicized.

28. Line 691. Include the names of the analyzed genes, as they are not listed in the Methods section.

Response: We thank the reviewer for this suggestion. The names of the analyzed genes and their associated functions are provided in Figure S9, where they are listed in detail. We have clarified this in the Methods section to explicitly direct readers to Figure S9 for this information.

29. Line 350. Italicize the gene names: spa and mec from SCCmec. Correct throughout the text.

Response: They were corrected throughout the manuscript.

30. Line 381. Discuss the MRSA/MSSA relationship and explain the difference.

Response: The MRSA/MSSA relationship observed in this study is largely driven by differences in hospital specialization and patient demographics. The collection includes isolates from three tertiary-care hospitals with distinct clinical profiles: the National Institute of Perinatology (INPer), which predominantly serves neonatal and pediatric

patients, and two adult-care hospitals (HDF and INC). As a result, MSSA predominated among pediatric isolates from INPer, where oxacillin resistance was relatively infrequent, whereas MRSA was more prevalent in adult-care settings.

The inclusion of a large pediatric cohort also allowed the identification of both transient expansions of MRSA lineages (e.g., CC8) and a high diversity of MSSA lineages (e.g., CC5 and CC30). We have revised the Discussion (l. 435-443) to clarify that differences in MRSA/MSSA proportions primarily reflect patient population structure and hospital specialization rather than sampling bias.

31. Lines 391-404. Discuss the possible presence of CA-MRSA strains in hospitals.

Response: Thank you for drawing attention to this point. We agree that the presence of community-associated MRSA (CA-MRSA) strains within hospital settings is plausible. In our collection, two isolates (INPER464 and INC065) display genomic features characteristic of CA-MRSA, including a complete SCCmec IV cassette carrying the COMER element and genes encoding Panton–Valentine leukocidin (PVL). These isolates were recovered from skin abscesses in adult patients. Other isolates may be of CA-MSSA origin, but we did not have clear evidence of them.

32. Lines 405-414. Discuss the types of strains found associated with the infections from which the strains were isolated.

Response: Thank you for this comment. We examined the distribution of strain types across infection sources, as summarized in Figure 2 (ring 3). Overall, no strong or consistent association was observed between specific strain types and particular infection sources. Both MRSA and MSSA isolates were recovered from a broad range of infections, including bloodstream, skin and soft tissue, and other clinical sources, suggesting that multiple lineages contribute to diverse infection types.

This lack of a clear association likely reflects the opportunistic nature of *S. aureus*, as well as the heterogeneity of the clinical sample represented in this study. We incorporated this paragraph to the Discussion (l. 406-412)

33. Lines 416-418. Include a supplementary table listing the accessory genomes found in the strains.

Response: a supplementary table was included with the description of the accessory genome and the COGs (Supplementary Table S8).

34. Lines 434-436. The fact that a strain possesses the *mecA* gene does not mean it is MRSA, as it can be a strain with a MIC <4 µg/mL that possesses the gene but is MSSA. MSSA

Response. Thanks for this reflexive observation, also in relation with question #10. All isolates were tested for oxacillin resistance and the presence of *mecA* with PCR. Oxacillin resistance was tested according to the CLSI standards as stated in Material and Methods section. In our experiments we consider the minimal inhibitory concentration (MIC) of 5 µg/mL of oxacillin to be considered as “oxacillin-resistant” isolate (Figure 1A). Further characterization of the isolates for the presence of *mecA* reinforces the classification of the isolates as MRSA. This preliminary screening resulted in 68/286 isolates resistant to oxacillin; 50 of them were positive for the presence of the *mecA* gene (Table S1). The rest did not show *mecA* presence. In the Figure 1A, only the isolates with the oxacillin resistant phenotype were considered. The genome of 38 out 39 isolates that were sequenced presented a phenotype of oxacillin resistance and presence of the *mecA* gene.

All isolates were confirmed first as *S. aureus* using standard microbiological tests (Gram staining, mannitol fermentation, including catalase, coagulase enzymes and resistance to cefotixin and oxacillin) and molecular methods, including PCR of the *coa* and *mecA* gene.

35. In the references, use italics for scientific names.

Response: All references were reviewed and scientific names were italicized.

36. Review the references to ensure they conform to the Author Guidelines and standardize them.

Response: References were formatted according to the ASM reference guidelines.

Re: Spectrum03140-25R1 (**Comparative Genomics of Methicillin-Resistant and -Susceptible *Staphylococcus aureus* from México City Hospitals**)

Dear Dr. Victor Gonzalez:

Your manuscript has been accepted, and I am forwarding it to the ASM production staff for publication. Your paper will first be checked to make sure all elements meet the technical requirements. ASM staff will contact you if anything needs to be revised before copyediting and production can begin. Otherwise, you will be notified when your proofs are ready to be viewed.

Sincerely,
Ayush Kumar
Editor
Microbiology Spectrum